# Adversarial Sampling for Solving Differential Equations with Neural Networks

**Kshitij Parwani**
Department of Mathematical Sciences
Indian Institute of Technology, Varanasi
`kshitijparwani.mat18@iitbhu.ac.in`

**Pavlos Protopapas**
John A. Paulson School of Engineering and Applied Sciences, Harvard University
Cambridge, Massachusetts 02138, United States
`pavlos@seas.harvard.edu`

## Abstract

Neural network-based methods for solving differential equations have been gaining traction. They work by improving the differential equation residuals of a neural network on a sample of points in each iteration. However, most of them employ standard sampling schemes like uniform or perturbing equally spaced points. We present a novel sampling scheme which samples points adversarially to maximize the loss of the current solution estimate. A sampler architecture is described along with the loss terms used for training. Finally, we demonstrate that this scheme outperforms pre-existing schemes by comparing both on a number of problems.

## 1   Introduction

Differential equations are ubiquitous in all engineering and science disciplines. Hence, substantial research goes into designing novel methods and improving pre-existing methods to solve differential equations. With the rise of deep learning, one approach which has gained traction is to use neural networks to solve these equations in an unsupervised manner. This has been used to solve a variety of differential equations such as ordinary differential equations (ODEs) [1, 2, 3, 4], partial differential equations [4, 5, 6, 7] (PDEs), and eigenvalue problems [8]. This holds certain advantages as opposed to numerical methods like finite difference such as (a) instead of obtaining solution values at discretized points, we get a closed and differentiable solution function [1], (b) it is more effective in solving high dimensional PDEs by faring better against the "curse of dimensionality" [5], (c) numerical errors are not accumulated in each iteration [4], (d) initial and boundary conditions are satisfied by construction [1, 2].

One method employed is to use a neural network (possibly in a reparametrized form to satisfy the initial/boundary conditions) to represent a candidate solution and minimize the loss corresponding to the differential equation over a number of iterations [1, 7]. This is done by sampling points over the domain using some predefined scheme (for example, uniform/equally spaced) and minimizing the loss function on these points in each iteration. However, such sampling schemes are either blind to the equation being solved or require one to carefully design custom sampling schemes to improve efficiency. We propose a novel method which samples points in an adversarial manner by finding points in the domain at which the current loss is high. This is followed by the solver minimizing the loss at these very points by updating the weights of the solver network. We demonstrate the efficiency of this scheme on a variety of problems.

DLDE Workshop in the 35th Conference on Neural Information Processing Systems (NeurIPS 2021).

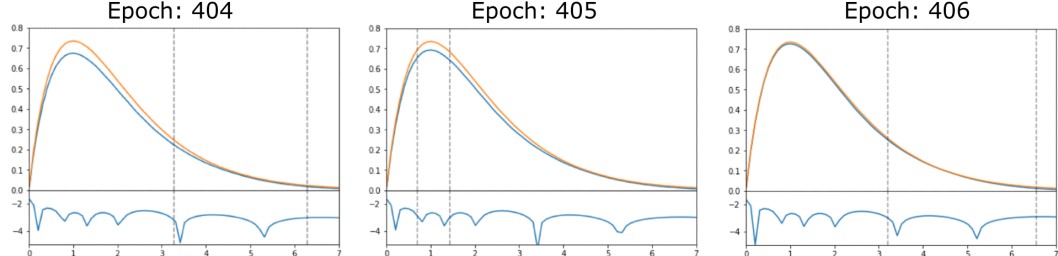

Figure 1: Solving an ODE with adversarial sampling, three consecutive iterations of the fitting process are shown on a small part of the input domain. It is observed that the sampler is able to steer points towards areas of high error and improve estimates. [**orange**] - analytic solution, [**blue**] - predicted solution, [**dashed**] - samples, [**bottom panel**] - error residuals (log scale)

## 2 Methodology

We first describe the vanilla method and then talk about adversarial sampling.

### 2.1 Vanilla Approach

Although the framework described is readily extendable to harder ODEs/PDEs (as demonstrated in the results 2) , in this section we consider a first-order ordinary differential equation $\mathcal{F}$ to be solved for $x \in [a, b]$:

$$\mathcal{F}(x, y, y') = 0 \tag{1}$$

given the initial condition $y(x_0) = y_0$. Let the function learned by the neural network be $y_N$. The method employed to exactly impose the initial conditions is to reparameterize $y_N$ to get $\hat{y}$ as [1, 7]:

$$\hat{y}(x) = y_0 + (1 - e^{-(x-x_0)})y_N(x) \tag{2}$$

For a fixed $\hat{y}$, we may exactly calculate the (partial) derivatives using automatic differentiation. In further text, we write $\mathcal{F}(x, \hat{y}, \hat{y}')$ as just $\mathcal{F}(x, \hat{y})$. Hence for a given $\hat{y}$, we may express the loss function at a point $x$ as:

$$\mathcal{L}(\hat{y}, x) = \big(\mathcal{F}(x, \hat{y})\big)^2 \tag{3}$$

Hence, to solve the equation, we must minimize the expected loss :

$$\arg\min_y \left( \int_a^b \mathcal{L}(y, x)dx \right) \tag{4}$$

At each iteration, we minimize the empirical loss at points $\boldsymbol{x} = \{x_1, x_2, \ldots, x_n\}$:

$$\hat{\mathcal{L}}(\hat{y}, \boldsymbol{x}) = \sum_{i=1}^n \mathcal{L}(\hat{y}, x_i) \tag{5}$$

The routine way to sample points $x_1, x_2, \ldots, x_n$ is to use some pre-specified scheme like uniform/equally-spaced sampling and to update the parameters of our neural network $y_N$ in each iteration to improve our estimate $\hat{y}$.

### 2.2 Adversarial Sampling

Using a predefined sampling scheme has some clear drawbacks. First, it is agnostic to the equation being solved as well as our current estimate $\hat{y}$. An efficient sampling scheme is expected to take into account where our current estimate is incorrect and sample extra points from that region to converge faster. When fitting the 1D exponential decay equation ($u_x = -\lambda u$), for example, sampling equally both before and after the elbow would be inefficient. This is especially valid in cases where the number of sampled points is limited and insufficient to cover the entire space (increasingly valid in higher dimensions).

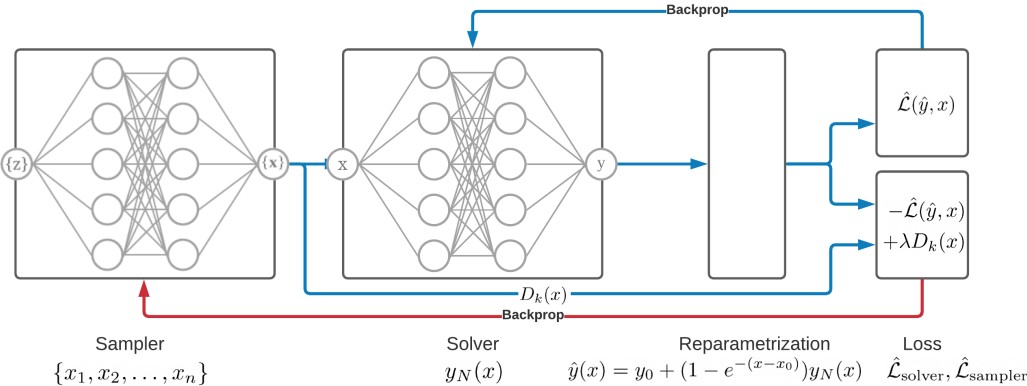

Figure 2: Architecture of the framework. The sampler network has input size $z_d$ (dimensionality of $z$) and output size $(n, d)$ ($n$ = number of samples and $d$ = dimensionality of each sample). The solver network has input size $d$ and output size of 1. It takes each $x_i \in \boldsymbol{x}$ and produces $y_N(x_i)$

We present a sampling scheme that is dependent on the current estimate $\hat{y}$. This is done by using a neural network to represent a variable sampling distribution implicitly similar to the generator network of a GAN [9]. In each iteration, the sampler is trained to produce points which maximize the loss of the solver (and a secondary loss). Thus, it competes with the solver whose weights are updated to minimize the loss at these very points.

### 2.3 Architecture of the Adversarial Sampler

We use a GAN-like [9] generator to produce samples. It is a fully connected neural network that uses a noise vector $z$ to produce a vector of sampled points $\boldsymbol{x}$. We use tanh activation in the last layer and rescale the output of the sampler to ensure the points lie in the specified input domain.

### 2.4 Regularization

It is observed that if the sampler is purely optimized with the objective of maximizing $\hat{\mathcal{L}}(\hat{y}, \boldsymbol{x})$ (residual loss corresponding to the DE at samples $\boldsymbol{x}$), it tends to collapse all samples to one single point of high loss. This is akin to the phenomenon of mode collapse in GANs [9]. Therefore it is essential to regularize the sampler by penalizing distributions with low entropy. For this, we use an additional loss term $D_k$. Given points $\{x_1, x_2, \ldots, x_n\}$, we define $d_k(x_i)$ to be the sum of distances of $x_i$ from its $k$ nearest neighbors. Hence for $\boldsymbol{x} = \{x_1, x_2, \ldots, x_n\}$ the secondary loss term is defined as follows:

$$D_k(\boldsymbol{x}) = -\sum_{i=1}^{n} d_k(x_i) \tag{6}$$

This is implemented using a kd-tree [10], which is queried for each point to get its $k$-nearest neighbors in order to calculate $D_k$. Other methods of enforcing entropy constraints were experimented with but this proved to be faster and more effective in practice.

### 2.5 Training iteration

In each training iteration, we first sample points from the sampler to get $\boldsymbol{x} = \{x_1, x_2, \ldots, x_n\}$. This leads us to two losses, namely $\hat{\mathcal{L}}_{\text{solver}}$ and $\hat{\mathcal{L}}_{\text{sampler}}$.

$$\hat{\mathcal{L}}_{\text{solver}} = \hat{\mathcal{L}}(\hat{y}, \boldsymbol{x}) \tag{7}$$

$$\hat{\mathcal{L}}_{\text{sampler}} = -\hat{\mathcal{L}}(\hat{y}, \boldsymbol{x}) + \lambda D_k(\boldsymbol{x}) \tag{8}$$

In one single training iteration, we first calculate $\hat{\mathcal{L}}_{\text{solver}}$ and update the parameters of $y_N$. Next we evaluate $\hat{\mathcal{L}}_{\text{sampler}}$ and update the parameters of the sampler.

# 3 Experiments

We demonstrate the results with a number of differential equations as shown in Table 1. Table 2 contains the results, where the comparison is done by first setting a target loss and maximum iterations. The maximum iterations are chosen in such a way so that both sampling schemes take equal time to arrive at them. Then by fixing the number of points, we compare the average time taken to arrive at the target loss or maximum iterations(whichever comes first). We also provide the average loss achieved, which might be higher than the target loss in case the maximum iterations are reached without achieving the target loss.

We evaluate the mean-squared error loss w.r.t. the analytic solution in the case of ODEs where this is available, and in the case of PDEs we use the validation loss which is the residual loss evaluated on a grid of $(32, 32)$ equally spaced points. All average data is reported over 10 trials.

| List of Differential Equations evaluated | | | |
|---|---|---|---|
| Equation Name | Differential Equation | Domain | Initial/Boundary Conditions |
| Exponential decay $(\gamma = -5)$ | $u_x = e^{\gamma x}$ | $[0, 30]$ | $u\|_{x=0} = 0.1$ |
| Logistic Equation $(\gamma = -1, M = 1)$ | $u_x = \gamma u(M - u)$ | $[0, 10]$ | $u\|_{x=0} = 0.7$ |
| Radial part of H-atom $(n = 1, l = 0)$ | $u_{xx} = 2\left(\frac{1}{2n^2} - \frac{1}{x} + \frac{l(l+1)}{2x^2}\right)u$ | $[0, 30]$ | $u\|_{x=0} = 0, u\|_{x=\infty} = 0$ |
| Radial part of H-atom $(n = 2, l = 0)$ | $u_{xx} = 2\left(\frac{1}{2n^2} - \frac{1}{x} + \frac{l(l+1)}{2x^2}\right)u$ | $[0, 30]$ | $u\|_{x=0} = 0, u\|_{x=\infty} = 0$ |
| Laplace Equation | $u_{xx} + u_{yy} = 0$ | $[0, 1] \times [0, 1]$ | $u\|_{x=0} = sin(y), u\|_{x=1} = 0$ $u\|_{y=0} = 0, u\|_{y=1} = 0$ |

Table 1: Table summarizing the problems.

| Results | | | | | | | |
|---|---|---|---|---|---|---|---|
| Equation Name | Num points | Loss type | Target loss | Avg time (NL) | Avg time (Adv) | Avg loss (NL) | Avg loss (Adv) |
| Exponential decay $(\lambda = -5)$ | 30 | MSE | $10^{-6}$ | $8.547s$ | $4.179s$ | $1.866 \times 10^{-6}$ | $9.740 \times 10^{-7}$ |
| Logistic Equation $(\gamma = -1, M = 1)$ | 20 | MSE | $10^{-6}$ | $16.215s$ | $3.486s$ | $7.018 \times 10^{-3}$ | $8.795 \times 10^{-7}$ |
| Radial part of H-atom $(n = 1, l = 0)$ | 30 | MSE | $10^{-4}$ | $11.433s$ | $7.995s$ | $3.673 \times 10^{-4}$ | $9.045 \times 10^{-5}$ |
| Radial part of H-atom $(n = 2, l = 0)$ | 30 | MSE | $10^{-4}$ | $11.660s$ | $7.385s$ | $6.508 \times 10^{-4}$ | $1.179 \times 10^{-4}$ |
| Laplace Equation | 256 | VAL | $10^{-4}$ | $426.14s$ | $360.41s$ | $1.300 \times 10^{-4}$ | $1.002 \times 10^{-4}$ |

Table 2: Table summarizing the results. (NL) - Noisy Linspace, (Adv) - Adversarial Sampling. Noisy Linspace (NL) perturbs equally-spaced points with a fixed standard deviation. It is observed to perform the best among vanilla sampling schemes.

# 4 Conclusion

In this paper, we show that adversarial sampling works in practice. It is able to improve the efficiency of fitting and leads to lower loss. It is also observed that in order to avoid collapse, we must regularize by penalizing sampling distributions with low entropy. Future work would involve exploring this technique to solve high-dimensional PDE(s) where it holds more promise.

## Acknowledgments and Disclosure of Funding

The authors would like to thank the anonymous reviewers for their valuable feedback on this manuscript.

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
