# OpenReview forum: "Adversarial Sampling for Solving Differential Equations with Neural Networks"
_NeurIPS.cc/2021/Workshop/DLDE — DLDE Workshop -- NeurIPS 2021 Poster_

### Official Review · Reviewer_QQMS · 2021-10-06
**Interesting idea but worth further investigation**

**Confidence:** 4

**Review:**

The submission proposes a GAN-like structure for adaptively sampling the residual evaluation points during training, which is numerically shown to be efficient and accurate for solving several low-dimensional differential equations via neural network-based methods.

Novelty & Significance:

Recently, the neural network-based methods have achieved great empirical success in solving differential equations, where the training data points are often fixed over the entire training process. As such, re-sampling the collocation points periodically is promising to accelerate convergence and improve the performance of trained models. The authors migrate a GAN-like algorithm for re-sampling the training data points at each iteration, which outperforms the (noisy) uniformly spaced points in solving several differential equations. The idea is interesting but worth further investigation.

Questions:
1) This paper ignores several lines of work which is competitive with the given algorithm, e.g., r-adaptivity in finite element analysis, “Hierarchical deep-learning neural networks: finite elements and beyond”, “Solving Allen-Cahn and Cahn-Hilliard Equations using the Adaptive Physics Informed Neural Networks”.
2) The neural network-based methods are usually good for interpolation but may not be suitable for extrapolation. The authors should report the generalization ability of trained networks for unseen time, initial or boundary conditions, etc.
3) The experiments seem rather simple and easy, more challenging experiments (e.g., high-dimensional or highly nonlinear equations) should be carried out to demonstrate the necessity of using a GAN-like structure. The baseline (noisy linspace) is also not good enough, the proposed method should be compared with more advanced methods such as r-adaptive finite elements.

Minor:
1) formula (4): should be $\hat{y}$
2) line 45: period is missing at the end of sentence “… at each iteration.”
3) Figure 1: only 2 sample points (dashed) are employed for minimizing the empirical loss in each iteration?


**Score:**

2: Borderline paper

---

### Official Review · Reviewer_3jqN · 2021-10-11
**Novel idea to sample training samples for PINN; yet needs more study**

**Confidence:** 3

**Review:**

The paper presents a GAN-type design of architecture to sample training points to train physics-inform neural networks adaptively. The proposed method is shown to perform better both computationally and accuracy-wise as compared to (noisy) equidistance sampling.


# Novetly of the Idea
The paper discusses an important aspect of the success of the physics-informed neural networks (PINN) to solve (partial) differential equations. That is, the choice of the sampling points to train networks. Very often, a uniform sampling with noise is employed to train PINN.

The author(s) have proposed an interesting GAN-type architecture with PINN to adaptively sample points for training networks. Precisely, the points are re-sampled using the adversarial network to train PINN to fulfill the desired goal. The efficiency of the method is illustrated by several examples and has shown better performance numerically compared to (noisy) uniform sampling.

# Problems
* The article fails the report and discuss the existing literature towards better sampling strategies to train PINN, e.g., see https://arxiv.org/abs/2007.04542, https://link.springer.com/article/10.1007/s00466-020-01928-9.
* Is the approach applicable to the cases where the domain is non-convex? Moreover, how does the approach extend, particularly adversarial network training?
* The approach requires the number of sample points $n$ as an input to the adversarial network. So, it appears that the adversarial networks aim at identifying good $n$ sample points to train PINN. However, the choice of $n$ still plays an important role. Can you please provide some guidelines to choose $n$?
* The experiment section needs more elaboration: (a) how is the target loss measured when the adversarial network is also present? (b) In the case of PDEs, $32\times32$ is too coarse a mesh grid to consider it as ground truth; thus, the loss can easily be misleading. So, it would be good to consider a finer mesh grid, e.g., $128\times 128$, (c) since the proposed method and noisy linspace use different points, how is avg loss computed?
* The details of the solver network are missing!
* Could you please state some details about training both networks (solver and adversarial)? Typically, GAN networks are challenging. Therefore, it would be good to discuss challenges in training networks if there are.
* Since there are already several advanced methods to sample training points as mentioned above, it is important to compare with these advanced methods to understand the performance of the method better.
* Would it be possible to consider conditional adversarial networks to sample only additional new points (and keep earlier chosen points)?

**Score:**

2: Borderline paper

---

### Official Review · Reviewer_uXFy · 2021-10-11
**Novel Methods for Adversarial Training for Differential Equations**

**Confidence:** 3

**Review:**

The authors propose the use of adversarial learning methods to control data-sampling, during the optimization procedure of a neural network solver, towards samples that give a higher error. To avoid mode collapse the neural network sampler (which acts as an adversarial to the solver) is regularized to predict high-entropy states. This is done efficiently by means of a KD-Tree that enables the authors to quickly query the nearest neighbors of a set of points and maximizing the sum of distances from each point to their neighbors. Experiments on several different differential equations show that this leads to better performance than doing uniform random sampling both in terms of final accuracy and speed of convergence.

Pros:
The paper is well written and the results are well presented but more results are needed to understand how sensitive this technique is to the task being considered.

The regularization technique proposed is novel but more details on how the choice of number of neighbors and lambda parameter are needed.

Cons:
The biggest limitation of the paper is that it lacks discussion of alternative sampling strategies to uniform sampling and discuss how the method proposed differs from these (whether it can be seen as complementary contribution or not).

There are many other methods to avoid mode-collapse - it would be interesting to see if these methods may add further benefit to the method proposed in this work.

Minor things:

Lambda is used in two different ways in the paper. It would be interesting to know more about the effect of having regularization vs. not and know what this hyper-parameter is set to for reproducing the experiment giving the best results.

**Score:**

3: Good paper

---

### Public Comment · (anonymous) · 2021-12-08
**Concerns about the similar ideas of employing adversarial sampling**

This work presents similar contributions as [1] (https://arxiv.org/abs/2104.14320), which also proposes to use a generator network to sample additional high-loss samples for improving PINN performance. The generator loss is formulated as the -(solver_loss) + (MSE between the generated samples and the initially sampled data points, to prevent the generated samples from being intensively clustered around a single point in the domain (avoiding the mode collapse in GAN)).

The main difference from [1] is that, in order to avoid the mode collapse, this paper prefers each of the generated points to be distant from its k-nearest neighbors?

Reference:

[1] Thanasutives, Pongpisit, Masayuki Numao, and Ken-ichi Fukui. "Adversarial Multi-task Learning Enhanced Physics-informed Neural Networks for Solving Partial Differential Equations." 2021 International Joint Conference on Neural Networks (IJCNN). IEEE, 2021.

---

### Decision · Program_Chairs · 2021-10-17

**Decision:**

Accept (Poster)

**Comment:**

This paper represents a novel contribution to the theory of learning solutions to differential equations with deep neural networks using a novel min-max game theoretic objective that is shown to offer benefits to uniform sampling approaches. The work has been selected as a poster for the workshop. The authors may consider this opportunity to address limitations of the work that has been flagged by reviewers.